# The Relevance of Fetal Abdominal Subcutaneous Tissue Recording in Predicting Perinatal Outcome of GDM Pregnancies: A Retrospective Study

**DOI:** 10.3390/jcm9103375

**Published:** 2020-10-21

**Authors:** Friederike Weschenfelder, Nadin Baum, Thomas Lehmann, Ekkehard Schleußner, Tanja Groten

**Affiliations:** 1Department of Obstetrics, University Hospital Jena, 07747 Jena, Germany; friederike.weschenfelder@med.uni-jena.de (F.W.); nadinbaum@t-online.de (N.B.); ekkehard.schleussner@med.uni-jena.de (E.S.); 2Institute of Medical Statistics and Computer Science, University Hospital Jena, Friedrich Schiller University Jena, 07747 Jena, Germany; thomas.lehmann@med.uni-jena.de

**Keywords:** gestational diabetes, fetal abdominal subcutaneous tissue, estimated fetal weight, perinatal complications, large for gestational age, ultrasound predictors

## Abstract

Guidelines on the management of gestational diabetes (GDM) instruct physicians to involve ultrasound-based monitoring of fetal growth in addition to blood glucose. So far, glucose control besides clinical parameters like maternal body mass index (BMI) and gestational weight gain have been shown to predict neonatal outcome. We aimed to evaluate the discriminative ability of fetal abdominal subcutaneous tissue (FAST) in addition to standard ultrasound parameters like abdominal circumference (AC) and estimated fetal weight (EFW) for perinatal complications like large for gestational age (LGA), hypoglycemia, hyperbilirubinemia, mode of delivery and admission to neonatal intensive care unit (NICU). Ultrasound data and neonatal outcome was collected of 805 GDM cases from 2012 to 2016: 3205 FAST, 3195 AC-measurements and 3190 EFW calculations were included. AC, EFW and FAST increased linear with gestational age. Combining ultrasound and clinical parameters improved predictive power for LGA. In the subgroup where fetuses grow with an AC > 75th additional adding of FAST to standard ultrasound parameters increased predictive power for hypoglycemia. Our results confirm inclusion of ultrasound parameters to be beneficial in monitoring GDM pregnancies. Additional FAST determination revealed to be of potential clinical relevance in the subgroup AC > 75th percentile.

## 1. Introduction

During the last couple of years Ultrasound (US) based monitoring of fetal growth has become more relevant for optimizing treatment of pregnancies affected by gestational diabetes mellitus (GDM). Adverse perinatal outcomes like newborns being born large for gestational age (LGA), necessity of C-section, neonatal hypoglycemia, hyperbilirubinemia and admission to neonatal intensive care unit (NICU) are still more frequent in pregnancies complicated by GDM [1,2,3,4,5,6]. The aim of GDM treatment is the reduction of these adverse outcomes to frequencies observed in non-GDM pregnancies.

Ultrasound based monitoring of fetal growth was established to identify fetuses at risk for overgrowth and associated impaired perinatal outcome to enable physicians to adapt treatment in these cases. German guidelines recommend to involve fetal growth-parameters into the treatment of GDM patients, focusing on fetal abdominal circumference (AC) and estimated fetal weight (EFW) [7]. Recently, several studies on fetal abdominal subcutaneous tissue (FAST) claiming this US parameter to be another noninvasive marker sensitive for the nutritional state of the fetus [8,9,10]. Most of these studies focused on the value of FAST in predicting fetal birth weight and especially LGA newborns. Madendag et al. showed that FAST measurements are able to predict LGA newborns with similar performance like AC and EFW [11].

In 2019, Rao et al. [12] published a review on the fetal biometry for guiding the medical management of women with GDM for improving maternal and perinatal health. They stated that there is insufficient evidence to evaluate the value of fetal biometry, in addition to maternal blood glucose values, to assist in guiding the medical management of GDM and the authors requested further studies on this particular subject [12].

Thus, the aim of our study was to confirm the positive predictive value of ultrasound parameters of fetal growth for diverse perinatal outcome parameters and to evaluate the extra benefit of using FAST in addition to AC and EFW in monitoring critical GDM subgroups.

## 2. Methods

This was a retrospective cohort study of GDM pregnancies, who were managed in the center for diabetes and pregnancy at the Department of Obstetrics at the Jena University hospital. Mother infant-dyads attending between 1st January 2012 and 31st December of 2016 were included. Exclusion criteria for this primary database were nonsingleton pregnancies and patients with pre-existing diabetes mellitus. GDM diagnosis was based on the results of a 75 g oral glucose tolerance test (oGTT) according to the IADPSG and WHO-2013 criteria [13,14]. According to the German pregnancy care guidelines, screening for GDM applying a 50 g glucose challenge test has to be offered to each pregnant woman. In cases of pathological results, a 75 g oGTT is mandatory. Diabetes care was provided according to the German S3 guidelines published in 2011 [15]. The local Ethical Committee of the Friedrich-Schiller-University, Jena, Germany approved this study (5280-09/17).

### 2.1. Data Collection

Clinical data collected included maternal age, parity, body mass index (BMI), treatment regime, gestational weight gain and data on glycemic control (HbA1c). Gestational age was calculated using the last menstrual period or earliest ultrasound. Body mass index (BMI) was calculated from maternal height and prepregnancy weight and gestational weight gain (GWG) as the difference of the prepregnancy weight and the last documented weight before delivery. Serum HbA1c levels were determined on a regular four-weekly basis and the mean of all HbA1c levels was calculated and used for statistical analysis in addition to the latest HbA1c obtained before delivery. Mean blood glucose (MBG) was calculated as the mean of all the patient’s 6-point self-monitored blood glucose profiles of the day prior to their regular consultations. Perinatal outcome data included mode of delivery, birth weight, gestational age at delivery, sex, Apgar score, postnatal NICU admission and neonatal hypoglycemia and hyperbilirubinemia. Sources of perinatal outcome data were standardized nationwide used perinatal documentation systems of our University hospital or patient’s maternity records. LGA was defined as fetal growth above 90th percentile and SGA as fetal growth below the 10th percentile using Voigt’s percentiles for the body measurement of newborns [16].

### 2.2. Ultrasound Parameters

Transabdominal ultrasound scanning was performed at most fortnightly and at least every fourth week during GDM treatment, leading to an average number of 3 scans in each case (min 1; max 8). Scanning was performed in our outpatient clinic by a constant group of clinicians using a 3–5 MHz transabdominal transducer on a Voluson GE 8 (GE Healthcare GmbH, Solingen, Germany). Fetal biometry including measurements of biparietal diameter (BPD), frontal occipital diameter (FOD), head circumference (HC), abdominal sagittal diameter (ASD), abdominal transverse diameter (ATD), abdominal circumference (AC) and femur length (FL) were performed using standardized anatomic views according to The International Society of Ultrasound in Obstetrics & Gynecology (ISUOG) guidelines [17]. FAST measurements were taken from the anterior abdominal wall in mm at the standard level of the AC. Figure 1 shows the measurement of the inner to the outer line of the echogenic subcutaneous fat layer [18]. In the rare cases of insufficient ultrasound conditions, FAST measurement was postponed to the following appointment. Images and measurements were processed by a computerized database (Viewpoint, GE Healthcare, Solingen, Germany), calculating percentiles using Hadlock’s formula for EFW [19].

We did not perform an inter observer test since measuring FAST is a routine measurement in our specialized outpatient clinic and a good inter observer correlation of FAST measurements has been proven by many other researchers [20,21].

### 2.3. Statistical Analysis

We included all patients that met inclusion criteria into our analysis. No prior sample size estimation was performed. Generalized linear mixed models with logit link function were fitted for the primary outcome parameters with clinical standard parameters (SP) and ultrasound parameters to estimate the predicted probability of the outcome. The variable “Clinical Standard Parameters” (SP) represents the combination of HbA1c level at delivery, mean HbA1c during treatment, prepregnancy BMI and GWG in the statistic model.

Receiver operator characteristics (ROC) analyses were performed to discriminate between patients with and without an event (e.g., LGA) by the predicted probabilities of the corresponding model. area under the ROC curve (AUC) with 95% confidence intervals were calculated to evaluate the accuracy of the prediction. DeLong test was applied to compare two ROC curves of different parameter sets [22]. Optimal cut-off regarding the prediction model was determined by Youden index criteria and sensitivity, specificity as well as positive and negative predictive values are reported [23]. Due to the combination of multiple items in the prediction model, we decided not to present the specific cut-off values because they have no clinical meaning. DeLong test and Youden index were used in statistical relevant cases only when the 95% AUC confidence intervals of the combined parameters and of the SP did not overlap. Statistical analysis was performed using SPSS 24.0 (IBM Corp. Released 2016. IBM SPSS Statistics for Windows, Version 24.0. Armonk, NY, USA: IBM Corp). A *p*-value < 0.05 was considered statistically significant.

### 2.4. Outcome Parameters

The primary goal of this study was to evaluate the predictive power of FAST, AC and EFW for LGA status of the newborn, postnatal hypoglycemia, C-section, NICU admission and hyperbilirubinemia. Predictive power of the US parameters should be compared to the predictive power of clinical standard parameters (SP; including HbA1c level at delivery, mean HbA1c during treatment, prepregnancy BMI and GWG). Secondary outcome of the study is to compare the predictive ability of US parameters for perinatal outcome data in pregnancies with abnormal fetal growth presenting with an AC above the 75th percentile or an intrauterine over nutrition and those presenting with an EFW below the 10th percentile potentially being growth retarded.

## 3. Results

Eight hundred and five GDM patients with singleton pregnancies were enrolled. Five cases were excluded due to missing data concerning FAST. Overall, 3222 measurements of FAST were documented. One implausible FAST > 8 mm was excluded and so were 16 measurements before 20 weeks of gestation. Finally, 800 pregnancies with a total of 3205 FAST and 3195 AC measurements and 3190 EFW calculations were included in statistical analysis (Figure 2).

### 3.1. Patient Characteristics

Patient characteristics are shown in Table 1. Median age was 31 years (interquartile range (IQR ) 28–35), gestational age at GDM diagnosis was 26.3 weeks (24.7–28.0), prepregnancy BMI was of 26.3 kg/m^2^ (IQR 22.9–31.6) and GWG was 12 kg (IQR 8.2–16.5). Furthermore, 44.3% of the cohort showed excessive weight gain according to the recommendations of the institute of medicine [24]. In addition, 335 patients (42.1%) were normal weight or underweight and 252 patients (31.8%) were obese and 42.5% of the women were insulin treated during their pregnancy. Median HbA1c at the time of delivery was 5.5% (IQR 5.2–5.7), total overall mean HbA1c was 5.3% (IQR 5.1–5.5) and MBG was 5.8 mmol/L (5.6–6.1) revealing the study cohort to be well-controlled.

### 3.2. Perinatal Outcome

In total, 12.1% of the children were born LGA and 6.1% SGA; 31.7% of the patients had to undergo C-section. NICU admission was necessary in 11.2% of the cases and 4.5% of the newborns suffered hypoglycemia and 25.6% hyperbilirubinemia. Further information on perinatal outcome is presented in Table 2.

### 3.3. FAST

FAST measurements retrieved between 24 and 42 weeks’ revealed values starting at 2.1 mm at 24 weeks and increasing to 4.75 mm at 40 weeks (Table 3 and Appendix A). As a result, we found that the week of gestation nearly fits the 50th percentile of FAST times 10 in mm, which serves as a convenient prompt during daily patient care. EFW and AC showed a linear increase accordingly. FAST values did not differ depending on fetal sex.

### 3.4. Discrimination Ability of US Parameters for Perinatal Outcome

In Table 4, results of the ROC curve analysis are summarized. Upon addition of the ultrasound parameters FAST, AC or EFW to the combination of clinical standard parameters (SP; including HbA1c at delivery, mean HbA1c, GWG and prepregnancy BMI) the discrimination ability for LGA increased from AUC = 0.661 (CI 0.621–0.700) for SP only to AUC = 0.703 (CI 0.666–0.741) for SP + FAST to AUC = 0.804 (CI 0.773–0.835) for SP + AC or AUC = 0.804 (CI 0.774–0.834) for SP + EFW. None of the ultrasound parameters nor the combination of all three ultrasound parameters proved to increase accuracy in prediction of perinatal outcome parameters as NICU admission, hyperbilirubinemia, type of delivery and hypoglycemia (Table 4).

Figure 3 shows comparative receiver-operating-characteristics curves for clinical standard parameters without and in combination with the US parameters AC, EFW and FAST in prediction of neonatal outcome parameters. The highest and significant probability of correct prediction of LGA was achieved by using a combination of US parameters in addition to clinical standard parameters. Combining AC and EFW with SP already showed a significant discrimination for LGA (AUC 0.815; CI 0.786–0.844; *p* < 0.01 using deLong Test compared to SP only). The combination of all three US parameters to SP showed the highest accuracy (AUC = 0.816 CI 0.787–0.845; *p* < 0.001 using deLong Test compared to SP only) (Figure 3a). The optimal cut-off of this ROC analysis revealed a sensitivity of 65.4% with a specificity of 84.6%, and a positive predictive value of 35% and a negative predictive value of 95.1%. (Figure 3a) Predictive probability for NICU admission, hyperbilirubinemia and mode of delivery or hypoglycemia revealed to be low using ultrasound parameters (Table 4).

### 3.5. Subgroup Analysis

In the subgroup where fetuses grow with an AC above the 75th percentile, the discrimination ability for LGA could not be increased by combining the US parameters with clinical standard parameters (AUC = 0.716; CI 0.667–0.766 (SP alone) vs. AUC = 0.719; CI 0.670–0.768 (US combined with SP). Accuracy for predicting hyperbilirubinemia slightly increased from AUC 0.561 to AUC 0.613 by adding US parameters to SP (see Table 5).

Figure 3b shows the receiver-operating-characteristics curves for clinical standard parameters for prediction of hypoglycemia and hyperbilirubinemia. AUC increased from 0.800 to 0.894 combining US parameters and clinical parameters revealing a significant difference (*p* < 0.001, deLong Test). The optimal cut-off of this ROC analysis revealed a sensitivity of 100% with a specificity of 69.4% and a positive predictive value of 11.9% and a negative predictive value of 100%. No relevant changes were found for NICU admission (AUC = 0.799 to AUC = 0.810) and C-Section (AUC = 0.639 to AUC = 0.651) (Figure 3b and Table 5).

In the subgroup presenting with an EFW below the 10th percentile ROC curve analysis adding US parameters to clinical standard parameters did not improve accuracy for NICU admission (AUC = 0.695 to AUC = 0.703) or hyperbilirubinemia (AUC = 0.691 to AUC = 0.715). Further addition of FAST increased the AUC for C-section nonsignificantly from AUC = 0.768 to AUC = 0.827 (Figure 3c). ROC analysis for LGA and hypoglycemia could not be performed due to the low number of cases in this subgroup.

## 4. Discussion

Our study provides percentiles for FAST in a Caucasian cohort of well-controlled GDM pregnancies calculated from more than 3000 FAST measurements, retrieved from longitudinal measurements in 800 GDM pregnancies. As far as we know, our study provides the largest number of FAST values that has been published to date. Interestingly, in comparison to other studies, our cohort showed lower FAST values. In our cohort (*n* = 3205 FAST measurements) the 50th percentile was 2.9 mm at 30 wog, 3.4 mm at 33 wog and 3.7 mm at 36 wog. Higgins cohort of 125 diabetic pregnancies (*n* = 335 FAST measurements) showed 3.5 mm at 30 wog, 4.5 mm at 33 wog and 5.5 mm at 36 wog, Chen’s Chinese cohort of 755 healthy, normal weight singleton pregnancies showed as 50th percentile: 4.2 mm at 31 weeks, 4.3 mm at 33 wog and 4.2 mm at 36 wog [9,25]. One obvious reason for such discrepancy could be due to different ultrasound techniques for measuring FAST. However, we followed the method published by Bethune et al. in 2003 [18]. Furthermore, in our study the number of included FAST measurements was ten and four times higher, supporting our data being more profound. Aside from the obvious difference in ethnicity, another reason might be due to differences in the actual nutrient situation of the fetuses, since there is rare or no information about the level of glucose control in the study of Higgins. The Chinese population might innately have higher FAST due to Asian ethnicity.

We could show that the combination of the three sonographic parameters (AC, EFW and FAST) lead to a significant higher discriminative accuracy for LGA compared to each single parameter. Against our expectations but consistent with Khalifa et al. we could show that FAST itself is even less accurate compared to AC or EFW in predicting LGA [10]. However, the highest overall discrimination for LGA was achieved by using the combination of all three US parameters in addition to the clinical standard parameters (HbA1c level at delivery, mean HbA1c during treatment, prepregnancy BMI and GWG) significantly increasing AUC = 0.661 (CI 0.621–0.700) to AUC = 0.816 (CI 0.787–0.845; *p* < 0.001).

In congruency with Khalifa et al. who showed that FAST could not be used as a predictor for the mode of delivery [10], in our study, none of the US parameters itself nor the combination of all three US parameters significantly enhanced the discrimination accuracy for maternal (C-section) or perinatal outcome parameters other than LGA (NICU admission, hyperbilirubinemia and hypoglycemia). There were slight improvements of accuracy still leaving AUCs of all mentioned perinatal complications between 0.5 and 0.7, representing only low accuracy without relevance for clinical purposes [26].

Since several studies claimed FAST was affected by maternal blood glucose levels and fetal hyperinsulinemia, we assumed that FAST would be a valuable predictor for fetal hypoglycemia [27,28,29]. However, our findings conclude that neither clinical standard parameters nor the US parameters could discriminate for hypoglycemia in the entire GDM cohort. These findings match the result of Ulrich et al., which could not find a correlation between FAST and fetal insulin levels measured by amniocentesis in GDM pregnancies in the early third trimester [30]. Results of our study were possibly impacted by the high quality of glucose control in the study cohort, indicated by the low HbA1c and blood glucose levels (see Table 1) and the consecutive high proportion of normally grown fetuses (see Table 2).

When fetal growth is abnormal, management of GDM in pregnancies is still the subject of ongoing discussion. Some study groups recommend individualized glucose targets if fetal growth is above the 75th percentile to lower goals and less strict glucose control in cases of decreasing growth velocity [31,32]. However, there might be symmetric big babies from large parents for whom growing above the 75th percentile is just normal and on the other hand, growth retardation might not be counteracted by just fueling the fetus with glucose. Thus, we asked the question whether using US parameters were useful to predict outcome and guide therapy management in these subgroups. For prediction of hyperbilirubinemia we were able to show a slight improvement in the subgroup with an AC > 75th percentile (hyperbilirubinemia AUC = 0.561 and combined AUC = 0.613). An even more important finding is the significantly better accuracy for discrimination for postnatal hypoglycemia in this particular subgroup increasing AUC from 0.800 (for clinical parameters) to AUC = 0.894 upon addition of FAST to AC and EFW. Thus, in fetuses growing with an AU exceeding the 75th percentile, FAST measurement demonstrated to distinguish healthy big babies from those suffering from hyperinsulinemia. As such, FAST measurement will be helpful to clinicians to decide whether to adapt glucose goals or not.

For the subgroup with an EFW below the 10th percentile, adding fetal US parameters to clinical parameters revealed to improve the discrimination ability slightly for C-section by shifting the AUC 0.768 to AUC = 0.827. Nevertheless, AUC for this question remained the same using AC and EFW only, leaving FAST without additional benefit.

### Strength and Limitations

A strength of our study is the large number of fetal ultrasound parameters especially of FAST measurements included into our analysis. In addition, our cohort represents a group of well-controlled GDM pregnancies, as is confirmed by the mean HbA1c at delivery of 5.5% with a range from 5.2% to 5.7%. However, a confounder to be mentioned might be the rather high percentage of insulin treated pregnancies (42.5%). Due to frequent admission of complicated GDM cases to our specialized unit, the rate of insulin-treated pregnancies is high compared to general data for GDM in Germany, where the rate of insulin treatment is about 30% [33]. However, it has been shown that rather the achieved glucose control and not the course of therapy impact on perinatal outcome. Furthermore, the small number of SGA babies (6.1%) in our cohort disproves an overtreatment. Another limitation is that only HbA1c was included into statistical model representing blood glucose control of the patients. We decided not to include MBG since values are based on documented self-monitoring glucose only and therefore more susceptible to errors. Additionally, the retrospective design and the reliability on the electronic patient records are always associated with limitations.

## 5. Conclusions

Our results implicate that measurement of the EFW, AC and FAST are useful parameters to monitor treatment in patients with GDM and help to discriminate those who are going to be LGA at birth. Furthermore, additional monitoring of FAST in the subgroup with an AC above the 75th percentile improves the forecast of hypoglycemia and should be considered for lower glucose targets. Thus, we could not prove FAST to be of additional value in the routine management of uncomplicated GDM pregnancies, but in subgroups with abnormal fetal growth, FAST can be helpful to distinguish between fetuses who need intensified monitoring and treatment and those who do not. For such cases, this paper provides percentile curves and profound data which help to interpret FAST measurements in GDM pregnancies.

## Figures and Tables

**Figure 1 jcm-09-03375-f001:**
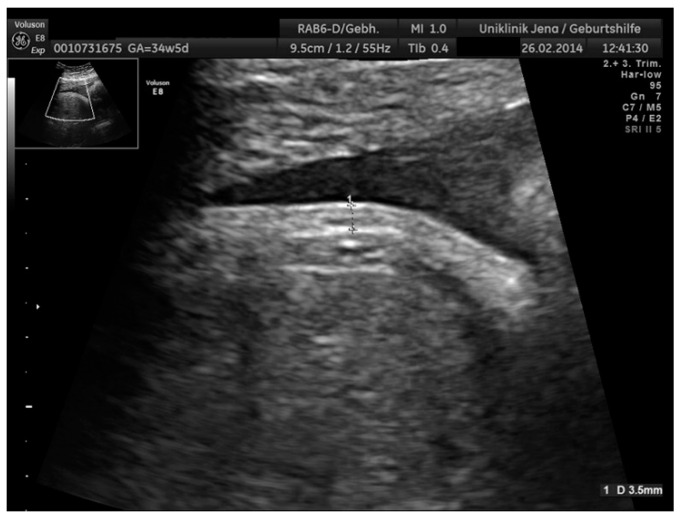
Ultrasound images showing fetal abdominal subcutaneous tissue (FAST) measurement of 3.5 mm at 35 weeks of gestation using Bethune’s method [18].

**Figure 2 jcm-09-03375-f002:**
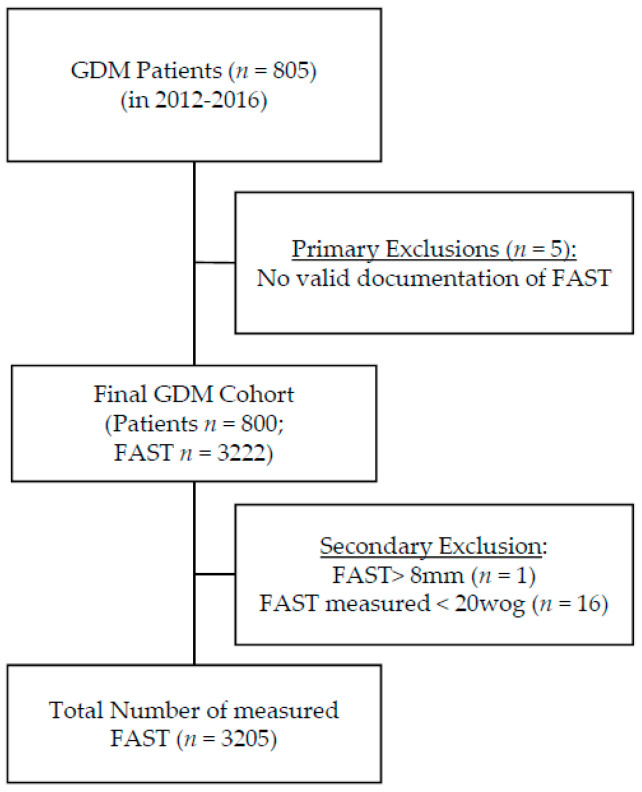
Cohort composition—the primary cohort consists out of 805 singleton GDM (gestational diabetes) pregnancies, supervised in our outpatient department for diabetes and pregnancy from 1st January 2012 to the 31st December of 2016. We excluded five mothers due to missing FAST (fetal abdominal subcutaneous tissue) values, one value was excluded due to implausible FAST >8 mm and 16 because measurements were derived before 20 weeks of gestation. Finally, 800 pregnancies with 3205 FAST measurements were included.

**Figure 3 jcm-09-03375-f003:**
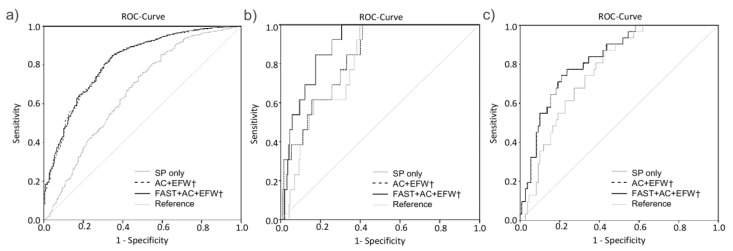
Receiver operating curves (ROC curves)for standard clinical parameters (SP) (

; light grey), standard clinical parameters combined with standard ultrasound parameters abdominal circumference (AC) and estimated fetal weight (EFW) (- - -; broken black line) and with additional inclusion of FAST (fetal abdominal subcutaneous tissue)measurement (

; black) in prediction of LGA in the entire cohort (**a**) in prediction of hypoglycemia in the subgroup of fetuses with an AC > 75th percentile (**b**) and for C-section when the fetus is growing below the 10th percentile (**c**). † means SP is included.

**Table 1 jcm-09-03375-t001:** Descriptive parameters of the entire cohort (*n* = 800).

Variable	Value
Maternal age in years (*n* = 800)	31 (28–35)
Parity (*n* = 782)	1 (0–1)
GA at GDM diagnosis in weeks (*n* = 698)	26.3 (24.7–28.0)
BMI in kg/m^2^ (*n* = 795)	26.3 (22.9–31.6)
BMI < 18.5 kg/m^2^ (underweight)	9 (1.1%)
BMI 18.5–24.9 kg/m^2^ (normal weight)	326 (41%)
BMI 25–29.9 kg/m^2^ (overweight)	208 (26.2%)
BMI 30–34.9 kg/m^2^ (obesity class I)	131 (16.5%)
BMI 35–39.9 kg/m^2^ (obesity class II)	84 (10.6%)
BMI ≥ 40 kg/m^2^ (obesity class III)	37 (4.7%)
GWG in kg (*n* = 761)	12 (8.2–16.5)
Excessive GWG (*n* = 761)	337 (44.3%)
Insulin Treatment (*n* = 800)	340 (42.5%)
Mean HbA1c levels in % (*n* = 767)	5.3 (5.1–5.5)
HbA1c at delivery in % (*n* = 622)	5.5 (5.2–5.7)
MBG (mmol/L) (*n* = 787)	5.8 (5.6–6.1)

Data are *n* (%) or median and interquartile range unless otherwise specified. BMI—body mass index; GA—gestational age; GDM—gestational diabetes; GWG—gestational weight gain; MBG—mean blood glucose.

**Table 2 jcm-09-03375-t002:** Perinatal outcome of 685 mother–infant dyads.

Variable	Value
GA at delivery in weeks (*n* = 683)	39 (38–40)
Birth weight in g (*n* = 685)	3440 (3127–3768)
Percentile at Birth (*n* = 667)	56 (30–77)
LGA (*n* = 667)	81 (12.1%)
SGA (*n* = 667)	41 (6.1%)
Male Sex (*n* = 675)	373 (55.3%)
Female Sex (*n* = 675)	302 (44.7%
Induction of Birth (*n* = 656)	259 (39.5%)
Spontaneous Delivery (*n* = 685)	440 (64.2%)
Vaginal Operative Delivery (*n* = 685)	28 (4.1%)
C-section (*n* = 685)	217 (31.7%)
Shoulder Dystocia (*n* = 279)	7 (2.5%)
pH (*n* = 644)	7.26 (7.20–7.31)
APGAR 5 min (*n* = 670)	9 (9–10)
NICU Admission (*n* = 624)	70 (11.2%)
Pre-eclampsia (*n* = 566)	39 (6.9%)
Jaundice (*n* = 560)	110 (19.6%)
Hyperbilirubinemia (*n* = 476)	122 (25.6%)
Phototherapy (*n* = 546)	32 (5.9%)
Hypoglycemia < 2 mmol/L (*n* = 402)	18 (4.5%)

Data are *n* (%) or median and interquartile range unless otherwise specified. APGAR—Score including Appearance, Pulse, Grimace, Activity, and Respiration; GA—gestational age; LGA—large for gestational age; NICU—neonatal intensive care unit; SGA—small for gestational age.

**Table 3 jcm-09-03375-t003:** FAST percentiles based on 3205 measurements of a Caucasian GDM cohort.

Weeks of Gestation	*n*	10th Percentile (in mm)	50th Percentile (in mm)	90th Percentile (in mm)
<24	70	1.50	2.00	2.89
24	57	1.48	2.10	2.72
25	95	1.76	2.30	3.00
26	136	1.80	2.30	3.00
27	196	1.90	2.50	3.10
28	189	1.90	2.50	3.30
29	244	2.10	2.70	3.50
30	247	2.10	2.90	3.60
31	262	2.40	3.10	3.90
32	261	2.40	3.20	4.00
33	252	2.60	3.40	4.40
34	267	2.60	3.40	4.50
35	217	2.70	3.60	4.90
36	269	2.90	3.70	4.90
37	184	3.05	3.90	5.05
38	176	3.00	3.90	5.30
39	64	3.15	4.20	5.10
40	15	1.00	3.80	5.00
>40	4	2.40	4.75	n.a.

Data shown in median only to provide better clarity. No significant differences for fetal sex. FAST—fetal abdominal subcutaneous tissue; n.a.: not available.

**Table 4 jcm-09-03375-t004:** Area under the receiver-operating curve for the perinatal outcome parameter: LGA, NICU admission, C-section, hyperbilirubinemia, admission and hyperglycemia using different combinations of clinical standard parameters and ultrasound-based predicting methods.

Outcome	Included Parameters	AUC	CI (95%)	*p*-Value ^‡^
LGA	SP only	0.661	0.621–0.700	<0.01
	FAST ^†^	0.703	0.666–0.741	<0.01
	AC ^†^	0.804	0.773–0.835	<0.01
	EFW ^†^	0.804	0.774–0.834	<0.01
	FAST + AC ^†^	0.806	0.776–0.836	<0.01
	FAST + EFW ^†^	0.805	0.775–0.835	<0.01
	AC + EFW ^†^	0.815 *	0.786–0.844	<0.01
	FAST + AC + EFW ^†^	0.816 *	0.787–0.845	<0.01
NICU	SP only	0.631	0.592-0.669	<0.01
	FAST ^†^	0.631	0.593–0.670	<0.01
	AC ^†^	0.637	0.598–0.676	<0.01
	EFW ^†^	0.640	0.601–0.678	<0.01
	FAST + AC ^†^	0.639	0.600–0.678	<0.01
	FAST + EFW ^†^	0.642	0.603–0.680	<0.01
	AC + EFW ^†^	0.639	0.601–0.678	<0.01
	FAST + AC + EFW ^†^	0.642	0.603–0.680	<0.01
C-section	SP only	0.599	0.574–0.625	<0.01
	FAST ^†^	0.600	0.574–0.625	<0.01
	AC ^†^	0.609	0.583–0.634	<0.01
	EFW ^†^	0.609	0.584–0.635	<0.01
	FAST + AC ^†^	0.609	0.583–0.634	<0.01
	FAST + EFW ^†^	0.609	0.583–0.635	<0.01
	AC + EFW ^†^	0.610	0.585–0.635	<0.01
	FAST + AC + EFW ^†^	0.610	0.585–0.636	<0.01
Hyperbilirubinemia	SP only	0.553	0.522–0.585	<0.01
	FAST ^†^	0.554	0.542–0.604	<0.01
	AC ^†^	0.573	0.542–0.604	<0.01
	EFW ^†^	0.566	0.536–0.597	<0.01
	FAST + AC ^†^	0.573	0.542–0.603	<0.01
	FAST + EFW ^†^	0.566	0.535–0.597	<0.01
	AC + EFW ^†^	0.573	0.535–0.597	<0.01
	FAST + AC + EFW ^†^	0.573	0.543–0.604	<0.01
Hypoglycemia	SP only	0.562	0.485–0.640	0.10
	FAST ^†^	0.567	0.494–0.639	0.08
	AC ^†^	0.563	0.485–0.640	0.10
	EFW ^†^	0.572	0.497–0.647	0.06
	FAST + AC ^†^	0.567	0.494–0.640	0.08
	FAST + EFW ^†^	0.573	0.502–0.643	0.08
	AC + EFW ^†^	0.572	0.502–0.643	0.06
	FAST + AC + EFW ^†^	0.575	0.507–0.643	<0.05

^†^ variables including SP; ^‡^
*p*-value presenting only the significance of AUC for the combination of variables in that specific row; *****
*p* < 0.001 using deLong test comparing to SP only; AUC—Area under the curve; LGA—large for gestational age; SP—Standard Parameters; FAST—fetal abdominal subcutaneous tissue; AC—abdominal circumference; EFW—estimated fetal weight; NICU—neonatal intensive care unit.

**Table 5 jcm-09-03375-t005:** Area under the receiver operating curve for perinatal outcome parameters (LGA, NICU, C-section, hyperbilirubinemia and hypoglycemia) in the subgroups with AC > 75th percentile or EFW < 10th percentile using combinations of clinical standard parameters and US parameters.

Subgroup	Outcome	Included Parameters	AUC	CI (95%)	*p*
AC > 75th Percentile	LGA	SP	0.716	0.667–0.766	<0.01
		AC + EFW ^†^	0.719	0.670–0.768	<0.01
		FAST + AC + EFW ^†^	0.719	0.670–0.768	<0.01
	NICU	SP	0.799	0.732–0.865	<0.01
		AC + EFW ^†^	0.810	0.745–0.875	<0.01
		FAST + AC + EFW ^†^	0.810	0.745–0.875	<0.01
	C-section	SP	0.639	0.594–0.685	<0.01
		AC + EFW ^†^	0.652	0.607–069.7	<0.01
		FAST + AC + EFW ^†^	0.651	0.606–0.697	<0.01
	Hyperbilirubinemia	SP	0.561	0.488–0.634	0.11
		AC + EFW ^†^	0.612	0.539–0.686	<0.01
		FAST + AC + EFW ^†^	0.613	0.539–0.686	<0.05
	Hypoglycemia	SP	0.800	0.720–0.881	<0.01
		AC + EFW ^†^	0.830	0.746–0.915	<0.01
		FAST + AC + EFW ^†^	0.894 *	0.838–0.949	<0.01
EFW < 10th Percentile	NICU	SP	0.695	0.609–0.782	<0.01
		AC + EFW ^†^	0.698	0.610–0.785	<0.01
		FAST + AC + EFW ^†^	0.703	0.614–0.792	<0.01
	C-section	SP	0.768	0.687–0.849	<0.01
		AC + EFW ^†^	0.827	0.753–0.901	<0.01
		FAST + AC + EFW ^†^	0.827	0.753–0.901	<0.01
	Hyperbilirubinemia	SP	0.691	0.556–0.826	<0.01
		AC + EFW ^†^	0.705	0.576–0.835	<0.01
		FAST + AC + EFW ^†^	0.715	0.588–0.842	<0.01

^†^ variables including SP; ***** —*p* < 0.001 using deLong test comparing to SP only; AUCs of Hypoglycemia and LGA are not presented in all subgroups due to missing cases. AUC—area under the curve; LGA—large for gestational age; NICU—neonatal intensive care unit; US—ultrasound; SP—Standard Parameters; FAST—fetal abdominal subcutaneous tissue; AC—abdominal circumference; EFW—estimated fetal weight; HC—head circumference.

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
