# Peer review of "The Relevance of Fetal Abdominal Subcutaneous Tissue Recording in Predicting Perinatal Outcome of GDM Pregnancies: A Retrospective Study"

_jcm, 2020, doi:10.3390/jcm9103375_

Round 1

Reviewer 1 Report

Thank you for giving me the opportunity to review this study. In this study, a relatively large number of patients has been included. However, due to the retrospective character of the study, data quality has to be questioned. The authors state that patients GDM diagnosis was based on IADPSG criteria. Was OGTT performed in all patients? At what gestational age? If not, what alternative diagnostic for GDM has been applied? The authors need to describe the exclusion criteria. Did the authors exclude patients with preexisting diabetes? Did the authors exclude multiple gestations (twins/triplets)?

At what gestational age did the investigators perform the ultrasound analysis? The authors have to be more clear about the exact methods here.

The paper needs to undergo extensive english language revision

Author Response

Point to Point response to reviewer 1:

Thank you for giving me the opportunity to review this study. In this study, a relatively large number of patients has been included. However, due to the retrospective character of the study, data quality has to be questioned. The authors state that patients GDM diagnosis was based on IADPSG criteria. Was OGTT performed in all patients? At what gestational age? If not, what alternative diagnostic for GDM has been applied? The authors need to describe the exclusion criteria. Did the authors exclude patients with preexisting diabetes? Did the authors exclude multiple gestations (twins/triplets)?

We thank the reviewer for this comment, of course, this information has to be included and we were very happy to do so. Please find our changes on page 2 line 58;59.  

According to German pregnancy guidelines screening for GDM is mandatory in Germany and the 50g GCT is mostly used for screening. However, to confirm diagnosis of GDM all patients have to underwent 75g oGTT in a two-step procedure. Thus, indeed, all patients received an 75g oGTT and GDM was diagnosed between 26 and 29 weeks of gestation. We also included this information in Table 1.

At what gestational age did the investigators perform the ultrasound analysis? The authors have to be more clear about the exact methods here.

Again, we thank the reviewer for this comment. Measurement of FAST was done at each visit of the patient at our outpatient unit. At least at a four-week schedule and at most at a two-week schedule. We clarified this procedure by revising Page 2 Line 82-84 in the method section.

The paper needs to undergo extensive english language revision

The manuscript was extensively revised for language editing by two native speakers.

Reviewer 2 Report

The authors  aimed   to evaluate the extra benefit of using  FAST in addition to AC and EFW in monitoring critical GDM subgroups concluding that additional FAST determination revealed to be of potential clinical  relevance in the subgroup AC> 75th percentile. 

The article  is well written albiet i would ask your comments for some remarks:

  1. Methods : a.Please clarify inclusion criteria; all diagnosed with GDM-actually cross sectional study -are all in the same stautus glucose,HAIC levels? b Wasn't there any influence of fetal position as back is  up by so  creating shadow over the area to measure. c Wasn't there any influence of obesity on quality of measuring FAST
  2. Since a cross sectional study you  do not have the information over contineous follow up are you limited in asseseing the added value of the FAST in managing and monitoring GDM women
  3. It is stated  that since the patients are monitored in a specialized clinic for GDM they are mostly monitored.  please share your thought related to the patients with AC>75  after all mode of delivery was the same when not using the FAST only EFW+AC   
  4. In table 4 the AUC is detailed for all parameters .One can see nearly  no difference compared to the other measured parameters. the clinical added value is not clear from your results .

Author Response

Point to point response to reviewer 2

The authors  aimed   to evaluate the extra benefit of using  FAST in addition to AC and EFW in monitoring critical GDM subgroups concluding that additional FAST determination revealed to be of potential clinical  relevance in the subgroup AC> 75th percentile. The article is well written albeit I would ask your comments for some remarks.

Methods : Please clarify inclusion criteria; all diagnosed with GDM-actually cross sectional study -are all in the same stautus glucose,HAIC levels?

We thank the reviewer for the comment. The information on inclusions/exclusion criteria definitely has to be included in the main text (So far it was only presented in Figure 1). Please find our changes on page 2 line 58;59.

To answer the second question we need to say that the approach of our study did not include cross-sectional design since ultrasound parameters, as well as the HbA1c and glucose levels analysed were retrieved longitudinal in all patients of our cohort of 800 GDM patients. To our understanding this design is longitudinal rather than cross-sectional. We tried to clarify the longitudinal design in the method section: page 2 Line 82-84.

Wasn't there any influence of fetal position as back is up by so creating shadow over the area to measure. and Wasn't there any influence of obesity on quality of measuring FAST?

Again, we thank the reviewer for this comment and added some more information on that topic in the main text. In cases of insufficient US conditions (f.e. dorsoanterior fetal position and/or obesity), the FAST measurement was postponed to the following visit. In Figure 1 it is shown that we needed to excluded five cases of the 805 GDM pregnancies due to missing FAST values. We changed the text accordingly page 3 line 91-93.  

Since a cross sectional study you do not have the information over contineous follow up are you limited in asseseing the added value of the FAST in managing and monitoring GDM women.

We thank the reviewer for this comment. As stated above indeed we did follow the individuals longitudinal during their pregnancies. Our study is based on a retrospective cohort analysis of the entire cohort of GDM patients who have been seen at multiple follow up appointments during their entire pregnancy. We added some more information and the average numbers of appointments in the text to clarify our approach. Average number of US scans/ appointments was 3 (min 1, max 8) as presented now. Page 2 Line 82-84.

It is stated that since the patients are monitored in a specialized clinic for GDM they are mostly monitored.  Please share your thought related to the patients with AC>75  after all mode of delivery was the same when not using the FAST only EFW+AC   

We thank the reviewer of this comment. As stated in the conclusions we did not see a benefit of using FAST except for predicting fetal hypoglycemia in the special subgroup of fetuses growing with an AU exceeding the 75th percentile. EFW and AC slightly improved prediction of the mode of delivery, however, without statistical difference compared to the standard clinical parameters. Adding FAST to EFW, SP and AC improved the prediction of Hypoglycemia – but only in the subgroup >75th percentile. These results indicate that we need to consider lower glucose targets this special subgroup to avoid fetal hypoglycemia and also to avoid treatment expansion in healthy big babies. We added some comments in the discussion to respond to this helpful comment. Page 10 line 263 to 271.

In table 4 the AUC is detailed for all parameters .One can see nearly no difference compared to the other measured parameters. the clinical added value is not clear from your results .

Again, we thank the review for this critical comment. Yes, there are almost no differences of AUCs in table 4. We were also surprised that none of the US parameters improved the prediction of the perinatal outcome parameters, except LGA. Nevertheless, we think that this is also a very important finding of our study. US seems to be less helpful for improving GDM treatment as expected in the general GDM cohort – but our findings implicate that it might be very useful in special subgroups like AC>75th percentile as stated above.

Round 2

Reviewer 1 Report

The authors sufficiently revised their manuscript according to the reviewers comments